# Effect of Intermediate Plus Vaccine and vvIBDV on Bursa Secretory Cells and Their Glycoprotein Production

**DOI:** 10.3390/v15061301

**Published:** 2023-05-31

**Authors:** Imre Oláh, Balázs Felföldi, Zsófia Benyeda, Nándor Nagy, Attila Magyar, Emőke Szőcs, Ádám Soós

**Affiliations:** 1Department of Anatomy, Histology and Embryology, Semmelweis University, 1094 Budapest, Hungary; nagy.nandor@semmelweis.hu (N.N.); magyar.attila@semmelweis.hu (A.M.); szocs.emoke@phd.semmelweis.hu (E.S.); soos.adam@phd.semmelweis.hu (Á.S.); 2Ceva-Phylaxia Ltd., 1107 Budapest, Hungary; balazs.felfoldi@ceva.com; 3Biovo Animal Health Ltd., 7700 Mohács, Hungary; info@prophyl.hu

**Keywords:** chicken, bursa of Fabricius, glycoprotein, microenvironment, effect of IBDV vaccination and infection

## Abstract

There are two types of secretory cells in the chicken bursa of Fabricius (BF): (a) interfollicular epithelial cells (IFE), and (b) bursal secretory dendritic cells (BSDC) in the medulla of bursal follicles. Both cells produce secretory granules, and the cells are highly susceptible to IBDV vaccination and infection. Before and during embryonic follicular bud formation, an electron-dense, scarlet-acid fuchsin positive substance emerges in the bursal lumen, the role of which is unknown. In IFE cells, IBDV infection may induce rapid granular discharge, and in several cells, peculiar granule formation, which suggests that the glycosylation of protein is injured in the Golgi complex. In control birds, the discharged BSDC granules appear in membrane-bound and subsequently solubilized, fine-flocculated forms. The solubilized, fine-flocculated substance is Movat-positive and can be a component of the medullary microenvironment, which prevents the medullary B lymphocytes from nascent apoptosis. Vaccination interferes with the solubilization of the membrane-bound substance, resulting in: (i) aggregation of a secreted substance around the BSDC, and (ii) solid lumps in the depleted medulla. The non-solubilized substance is possibly not “available” for B lymphocytes, resulting in apoptosis and immunosuppression. In IBDV infection, one part of the Movat-positive Mals fuse together to form a medullary, gp-containing “cyst”. The other part of Mals migrate into the cortex, recruiting granulocytes and initiating inflammation. During recovery the Movat-positive substance appears as solid, extracellular lumps between the cells of FAE and Mals. Possibly the Mals and Movat-positive extracellular lumps glide into the bursal lumen via FAE to eliminate cell detritus from the medulla.

## 1. Introduction

It is generally accepted, that the microenvironment (ME) of the primary or central lymphoid organs are responsible for the maturation of immunologically competent cells. In birds the primary lymphoid organs are the thymus and the bursa of Fabricius, which are responsible for T- and B-cell maturation, respectively. The functional unit of the bursa is a follicle, which consists of developing B-cells, epithelial reticular cells (ERC) of supporting network, bursal secretory dendritic cells (BSDC), macrophage-like cells (Mals) and follicle-associated surface epithelium (FAE) [1]. A few years after discovery of lymphoid dendritic cell in mammals [2], a bursa poietin was identified, which converted stem cells/prebursal cells to immunocompetent B-cells [3]. In 1977, we identified a secretory type cell in the medulla of bursal follicles. This cell produces secretory granules and two-three long processes with small spike-like evaginations [4,5]. The assumption is that the product of the secretory cell interacts with pre-bursal B-cells, therefore the product of the secretory cell is indispensable for B-cell maturation [6].

Although the concept of B-cell lineage was established in chicken [7], it became clear later that the human and mouse model is largely different. The most important differences are that in birds the process of gene rearrangement in pre-bursal B-cells occurs only in one wave during embryonic development [8], which is in contrast with humans and mice, where B-cells are continuously produced in the bone marrow during the life period. Further maturation takes place in a limited time frame at the late embryonic stage between 15 and 18 days of incubation, regulated by developmental signals that are not yet determined, which are provided by the bursal microenvironment [9]. In the bursa, B-cells are constantly dependent on microenvironmental signals that prevent instant apoptosis and initiate developmental steps [10]. Fully matured B-cells colonize the peripheral lymphoid organs in the first weeks of age and establish self-sufficient populations that provide the pool for humoral immunity throughout the lifetime [11]. The bursa is the main target organ of IBDV infection, affecting bursal secretory dendritic cells (BSDC) and B-cells, causing severe depletion of lymphoid follicles, leading to immunosuppression [12]. Previously we have shown that IBDV infection, in addition to damaging the cellular components, causes changes in the composition of the microenvironment [13] indicating the essential role of gp in the physiological and immunological function of BF.

Recently it was demonstrated that the medullary region of bursa contains a mucin-type glycoprotein, that appears to be associated with BSDC [14]. The role of mucins in immunological functions and cancer research was extensively investigated in humans, revealing that in the secondary lymphoid organs, the follicular dendritic cells (FDC) express cell membrane-bound mucin MUC-1, which is involved in the activation of T- and B-cells in the mucosal immune system [15]. The bursa gp is detectable by Movat pentachrome staining [16] in the extracellular space of follicular medulla and in the Mals of IBDV infected bird [13,17].

The surface epithelium of bursal lumen consists of 90% and 10% of interfollicular (IFE) and follicle-associated epithelium (FAE), respectively [18]. The cells of IFE produce mucin-like substance, but these epithelial cells are not goblet-shaped cells and not of endodermal, but ectodermal origin [19]. This paper summarizes the secretory function of BF in control, vaccinated and infected birds.

## 2. Materials and Methods

### 2.1. Animals and Embryos

Two types of embryos and animals were used: New Hampshire and White Leghorn layer type chickens of SPF status. Before setting, the fertilized New Hampshire eggs were stored for 12 h at room temperature. The White Leghorn chickens were reared for 4 weeks under SPF conditions at the Animal House of Ceva Phylaxia. After IBDV vaccination at 4 weeks of age, chickens were placed into isolator units and kept there until termination. Conditions were set according to the need of chickens: 26 °C ambient temperature; 150 mPa overpressure; 30–40 m^3^/h air exchange; 12–12 h light/dark cycle; food (sterilized commercial broiler grower feed) and water (tap water), available ad libitum; environmental enrichment provided by ladders and sand containers.

The project was recognized with the internal identification number of 5/2019, as part of the concession, approved by the National Scientific Ethical Committee on Animal Experimentation for animal trials with live Infectious Bursal Disease in chickens (#12/2017). The animals were kept under conditions according to the EU decree No. 40 of 2013. Euthanization and humane endpoint to moribund specimens was performed by intravenous anesthetics (sodium pentobarbital) at dosage of 120 mg/body weight New Hampshire embryos were killed at 8, 10, 12, 13, 14, and 16 days of incubation. White Leghorn layer type embryos were killed at 13, 14, 15 and 16 day of incubation.

Seven-week-old New Hampshire breed of chickens were injected intraperitoneally with cyclophosphamide (Cy) on five consecutive days. On day one, the chickens were injected with 50 mg Cy/kg body weight followed with 25 mg/kg body weight. The chickens were killed by pentobarbital injection. The bursas were fixed in 4% phosphate buffered glutaraldehyde, overnight. The procedure was followed by 1% buffered osmium tetroxide fixation. After dehydration in graded ethanol, the tissue samples were embedded in Polybed-Araldite mixture (Polysciences, Warrington, PA, USA). The semithin (1 µm thick) sections were stained with toluidine blue. The thin sections were contrasted by uranyl-acetate and lead citrate, studied with Hitachi H-7500 (Hitachi Ltd., Hitachi, Japan) and Jeol Jem 1200 EX electron microscope (JEOL Ltd., Akishima, Japan).

### 2.2. IBDV Strains

For vaccination, 228E attenuated classical IBDV strain was used at dosage of 3.0 lgEID50/chicken, diluted in 0.2 mL sterile PBS applied, individually per-os. Vaccination was performed at 28 days of age. Very virulent IBDV isolate D407/2/04/TR at dosage of 4.0 lgEID50/chicken, applied individually per-os, diluted in 0.2 mL sterile PBS. The chickens were killed by overdosed pentobarbital injection at 3-, 4- and 5-days post vaccination and post-infection (32–33 days of age). In non-vaccinated, infected group additional samplings were included at 36- and 48-h post-infection to monitor the progression of IBDV infection. The results reported have come from five different experiments.

### 2.3. Glycoprotein Demonstration

The tissues were fixed in 4% formalin and embedded in paraffin. The 6–7 µm-thick sections were stained with Russell modification of Movat pentachrome staining [16].

### 2.4. Image Processing

Images were processed using Adobe Photoshop.

## 3. Results and Discussion

Based on cytological structure, two kinds of secretory type cells are present in the post-hatch BF: IFE cells and BSDC. The embryonic epithelial and IFE cells secrete into the bursal lumen (Figure 1a–d,g,h) while BSDC discharges into the extracellular space of follicular medulla (Figure 1e,f). Both IFE cells and BSDC are highly sensitive for IBDV vaccination and infection.

### 3.1. Secretion of IFE Cell and Embryonic Epithelium

In control bursa, the Movat pentachrome staining shows mucin (gp) in the cells of IFE, but not in the FAE [13,17]. The secretory granules of IFE cells are transparent, showing a dotted, moderately electron-dense material (Figure 1g). Possibly, the transparent portion of granules consist of carbohydrate, while the dotted electron-dense material may represent the protein portion of granules. In the embryos, before formation of epithelial buds and FAE, the surface epithelium “secretes” or “exudates” an electron-dense substance (Figure 1b), which gradually “dilutes” towards the central part of bursal lumen (Figure 1a). This substance, after Movat staining, is not blue-green, like the mucin, but intense red with scarlet-acid fuchsin (Figure 1c,d). According to the data sheet, fibrinogen and fibrin give intense red staining. The actual biochemical content and role of this highly electron-dense, scarlet-acid fuchsin positive substance is unknown, but the fibrinogen can contribute to the follicle or FAE formation. Before formation of FAE and IFE the epithelial cells do not contain any secretory granules (Figure 1b), unlike the cells of IFE (Figure 1g).

In the vaccinated and infected birds, the IFE cells rapidly release large amounts of mucin, and the mucin production remarkably diminishes (Figure 1h). In some IFE cells, the formation of secretory granules may change (Figure 2c–f). The altered granules are not spherical, but conical-shaped (Figure 2e) and show semi concentrical, toothed, electron-dense, parallel lines, and between the lines there are electron-lucent “empty” spaces. The conical-shaped granules seem to be polarized. In the tip of the granules, the semi concentrical lines may form complete circles (Figure 2f). In these cells, among the lamellated granules, there are several, moderately electron-dense granules without electron-lucent areas (Figure 2f). Other IFE cells have highly packed small “mini” granules, which are translucent with one-two electron-dense dots (Figure 2d). In these cells, normal and lamellated granules also formed (Figure 2d).

The secretory granules of IFE cells come from the trans face of the Golgi complex. In the protein and carbohydrate assemblage, the glycosylation takes place when the protein from granular endoplasmic reticulum passes through the Golgi complex. Possibly, the unique or malformed secretory granules emerge in the cells, which are actually in the phase of glycosylation during infection. The moderately electron-dense and the “mini” granules may contain only protein (Figure 2f) and carbohydrate (Figure 2d), respectively. In the cells of the IFE, we were unable to identify virus particles with IBDV cytochemistry or transmission microscope; however, the presence of malformed granules shows that IBDV infection deeply interferes with the Golgi function, glycosylation. It is not apparent whether the Golgi complex has been influenced directly by the virus or by some biologically active factor(s) released from other infected cells, namely BSDC transformed to Mals [13,17] or heterophil granulocytes [20] which enter the IFE.

### 3.2. Secretion of Bursal Secretory Dendritic Cell (BSDC) and IBDV Susceptibility

In control birds, the discharged BSDC granules temporarily attach to the cell membrane of BSDC and appear in membrane-bound form [13]. The membrane-bound form of granular discharge slowly solubilizes and spreads in the extracellular space of the medulla. The solubilized gp appears as a fine-flocculated (Figure 3a), Movat-positive (Figure 1e) substance in the medulla. Several follicles show intense staining, while others have a very weak reaction or none, suggesting that the gp production is highly different among the follicles. Possibly, the secretory cycle of BSDC is not synchronized, i.e., in each follicle, the ratio of immature and mature BSDC could be different. Thus, the different “functional” stage of each follicle may be determined by the number of BSDC in the secretory phase.

These findings may be confirmed by cyclophosphamide (Cy) treatment of pullets [4,5]. Cy induced severe cell depletion and an abundant electron-dense substance in the extracellular space of the medulla of several follicles (Figure 1f). The Cy-resistant, pale-stained cells, with vesicle-like nucleus and few cytoplasmic granules could be modified BSDC embedded in the dark extracellular substance (Figure 1f). Occasionally, among the pale-stained cells, cells with a small lymphocyte-like nucleus also appear (Figure 1f), which could be the remaining precursors of BSDC located in the cortico-medullary epithelial arch (CMEA). Currently, we do not know the amount of extracellular gp that is associated with IBDV susceptibility, but we assume that follicles, which contain many mature BSDC(s) that produce large amount of extracellular substance, are less susceptible to IBDV infection, because the gp holds the chB6 apoptotic receptor and protects the B-cell from nascent apoptosis.

### 3.3. BSDC Product; Glycoprotein and Vaccination

There are a wide range of histological alterations between the acute and chronic phases of virus replication. In the acute phase of vaccination, the histological findings are more similar to that of a mild infection, without medullary “cyst” formation (see below). In the chronic phase of vaccination, intense secretion takes place in both the medulla by BSDC and in the bursal lumen by IFE cells (Figure 1h and Figure 2a).

In the chronic stage of vaccination, the fused BSDC granules embed the virus particles (Figure 4c) and prevent granular discharge. The earlier secreted granular substance is aggregated and stably attaches to the BSDC membrane, suggesting that the solubilization of membrane-bound gp may be inhibited (Figure 3b–d and Figure 4a,b). The aggregation of already solubilized gp results in large, patchy, sharply-outlined, solid material in the B-cell depleted medulla (Figure 3e and Figure 4d). The non-solubilized, membrane-bound and aggregated gp may be partially or completely unavailable for medullary B lymphocytes [20,21], which can cause long-lasting immunosuppression. [22,23]. It is surprising that in vaccinated birds, remarkable numbers of plasma cells emerge in the medulla. The terminal maturation of B-cells, which is possibly inhibited in the BF of normal pullets, results in plasma cells. The vaccination may suspend the inhibition of terminal maturation of B-cells resulting in emergence of plasma cells in the medulla (Figure 3e).

### 3.4. BSDC Product; Glycoprotein and vvIBDV Infection

In vvIBDV infected birds, the gp appears in the Mals (transformed BSDC) (Figure 2b); some of the Mals rapidly migrate into the cortex, while some remain in the medulla and fuse together creating a “cyst”, which is full of gp (Figure 5a–f). The gp comes from the virus-destroying Mals (Figure 5a–f). It has been described by others that virus infection can cause “cyst” formation in the medulla of bursal follicles [14,24]. The “cyst” is full of gp (Figure 5f), Mals with virus particles (Figure 6a), and disintegrated cells (Figure 6b). The size and content of the “cyst” gradually diminishes and finally, the “cyst” and CMEA collapse (Figure 5b–e). If a few lymphoid-like cells, precursors of BSDC, remain in the CMEA, the follicle may be recovered. Chicks produce two kinds of follicles during recovery of neonatal IBDV infection [25,26]. One of them is large with a distinct cortex and medulla, and the other is small with “absence of cortico-medullary structure”. The small follicles are unable to mount antibodies to IBDV or Salmonella enteritidis; thus, the B-cells in the small follicles are unable to produce an effective immunoglobulin repertoire. There is a strong functional connection between the cells of CMEA and the lymphoid-like cells (precursors of BSDC): Notch signaling [27]. In the small follicles, the lack of cortico-medullary structure results in absence of Notch signaling and functioning BSDC precursors. The lack of mature BSDC and its product, gp, may influence the immunoglobulin repertoire. The “wall” of the “cyst” consists of stellate-shaped epithelial reticular cells (ERC), immature dendritic cells, and Mals (Figure 6c–f). In the extracellular space, there are solid, highly electron-dense, Movat-positive lumps, some of them surrounded by a moderately aggregated substance (Figure 6c). Mal(s) with virus particles seem to be isolated by ERC (Figure 6e,f). It is surprising that during recovery at 21 dpi, morphological alterations seen exclusively in the FAE (Figure 7a) are similar to those inside the whole medulla after vaccination. There is also an appearance of extracellular bodies or lumps and the discharged BSDC granular substance firmly attaching to the cell membrane of BSDC or cells transforming from BSDC to Mal (Figure 7b). These alterations provide evidence that there is a slow, continuous fluid flow in the healthy bursal follicular medulla toward the FAE. Possibly, the Mal gets in the FAE passively by the fluid flow. In the case of IBDV infection, the virus containing Mal actively migrates into the cortex to recruit heterophil granulocytes and initiate inflammation [20]. The chronic stage of vaccination may be “repeated” in late recovery of infection. The IBDV susceptibility of follicles may be determined by the amount of extracellular gp; namely the actual number of mature, secretory phased BSDC. In early stage of infection, heavily infected and non-infected follicles occur side-by-side [13], similar to the intensely, moderately, or non-Movat-stained follicles (Figure 1e). The alteration of BSDC product, gp, during vaccination and IBDV infection may be of high priority in a bird’s immunology and infection.

## 4. Conclusions

A secretory product of IFE cells cleans up the bursal lumen and duct, while FAE eliminates the medullary dead cells and detritus. In control birds, the BSDC secretes gp, which solubilizes in the extracellular space of medulla, contributing to bursal microenvironment and preventing ligation of chB6 molecule and nascent apoptosis of medullary B-cells. These essential functions of gp, and its remarkable changes in IBDV infection and vaccination underline the significance of BSDC-produced gp in birds. The IBDV infection results in Movat-positive, gp-containing “cyst” formation in the medulla, while in the chronic phase of vaccination, the solubilization of membrane-bound gp is inhibited, which subsequently changes the bursal microenvironment. In the chronic phase of vaccination, the membrane-bound, aggregated gp may not be “available” for the medullary B lymphocytes, which may result in immunosuppression until functional restoration of the follicle.

The scattered findings of secretory function of BF clearly show that two types of secretory cells are found in the bursa. (i) The IFE cells produce mucin (although IFE cells did not get enough attention) and IBDV infection, remarkably, decreases the mucin production [17]. (ii) Cyclophosphamide (Cy) treatment resulted in, besides severe B-cell depletion, a large amount of an extracellular, electron-dense substance in the medulla of several follicles [4,24]. (iii) The transmission electron microscope revealed a dark, electron-dense substance in the lumen of the embryonic bursa before and during follicle formation (previously unpublished). Recent findings clearly show that the classical extracellular matrix (ECM), fibers, and ground substance) products of fibroblasts and mesenchymal reticular cells (MRC) are found only in the interfollicular connective tissue and follicular cortex. The follicular medulla and bursal lumen have secretory products of BSDC and IFE cells, respectively. After IBDV vaccination and infection, the fundamental changes in extracellular gp and mucin production result in gp and mucin that are highly valuable in immunological and physiological functions of BF. In the follicular medulla, the extracellular gp is the first recognized biochemical moiety of bursal ME.

## Figures and Tables

**Figure 1 viruses-15-01301-f001:**
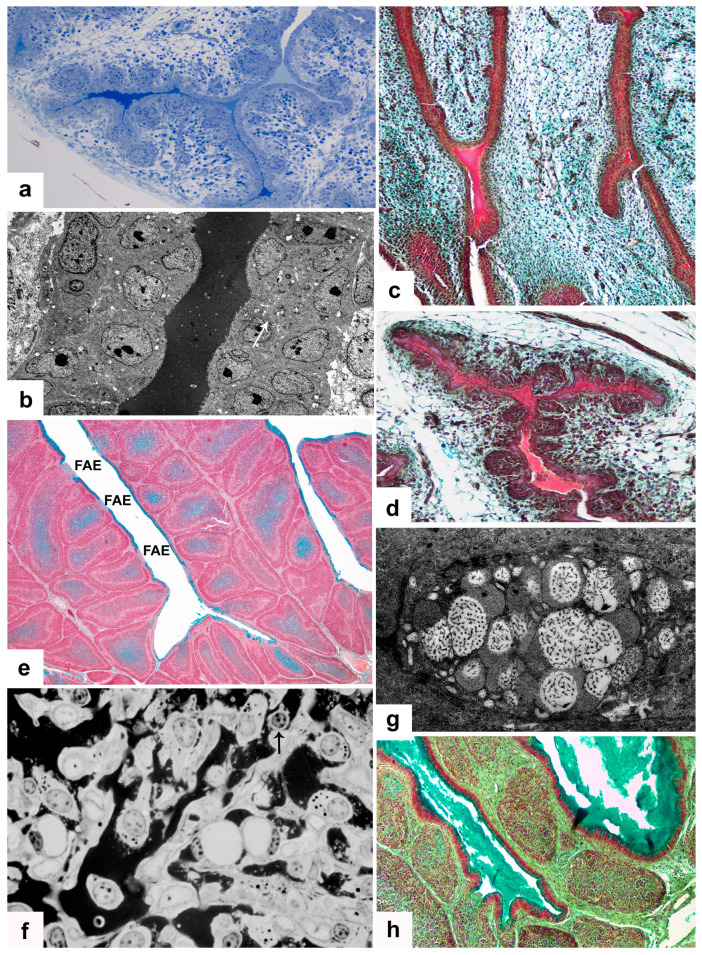
(**a**) 15-day old embryo. Semithin section (1.5 um thick), stained with toluidine blue, shows a homogenous material in the bursal lumen. (**b**) In 10-day old embryo, the transmission micrograph confirms the presence of an electron dense substance in the bursal lumen. (**c**) 12-day old embryonic bursa stained by Movat. In the bursal lumen, there is a scarlet acid fuchsin positive substance. (**d**) 15-day old embryo: Follicle formation stage. Scarlet acid fuchsin positive substance is still present in the bursal lumen. (**e**) BF from a five-week-old chicken, stained with Movat pentachrome staining. IFE cells are Movat-positive while the cells of FAE are negative. In the follicular medulla, the staining intensity is variable. (**f**) In several follicles, the Cy treatment of pullets resulted in an abundant dark extracellular substance. Cells with vesicle-like nucleus and pale-stained cytoplasm are embedded in a darkly stained substance in the medulla. Occasionally, the cell shows lymphocyte-like nucleus (arrow). (**g**) Mucin (gp) containing granules in the IFE cells of a pullet. (**h**) Three-days post-vaccination, the bursal lumen is full of Movat-positive gp.

**Figure 2 viruses-15-01301-f002:**
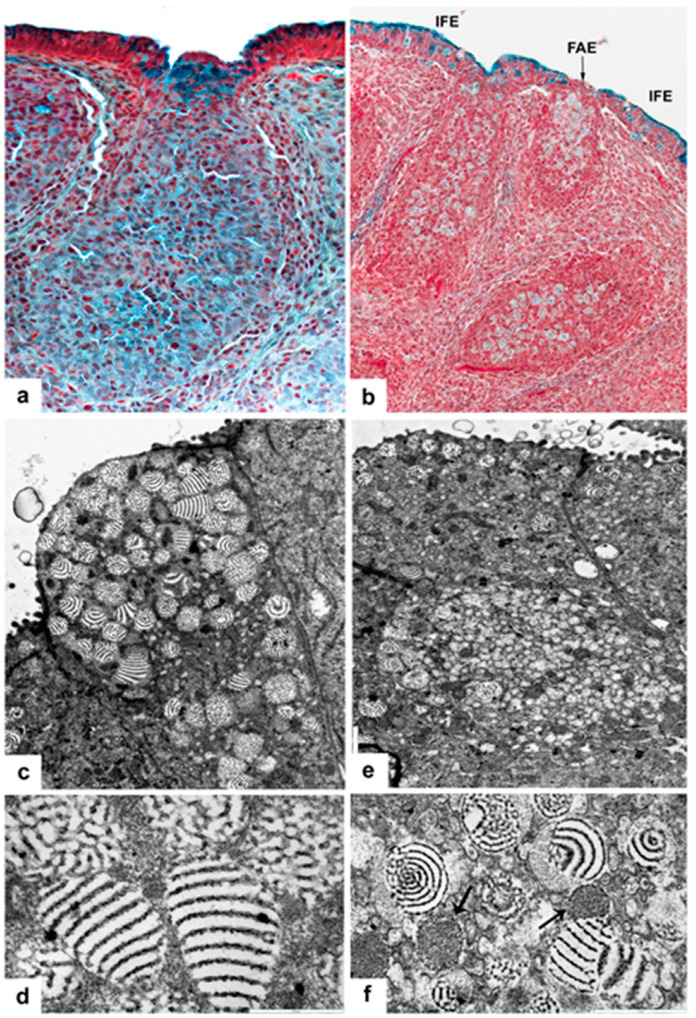
(**a**) Five days post-vaccination. The chronic stage of virus replication. The Movat staining shows remarkable amounts of extracellular gp and very intense gp bodies inside and below the FAE. (**b**) Five dpi, the gp appears intracellularly in virus containing Mal. (**c**) Five dpi several cells of IFE show unique granular structure. (**d**) Other IFE cells, besides the larger normal and lamellated granules, reveal very small secretory granules. (**e**) Higher magnification shows the conical-shaped, lamellated granules. (**f**) Among the lamellated granules, moderately electron-dense granules, without translucent area, also occur (arrow).

**Figure 3 viruses-15-01301-f003:**
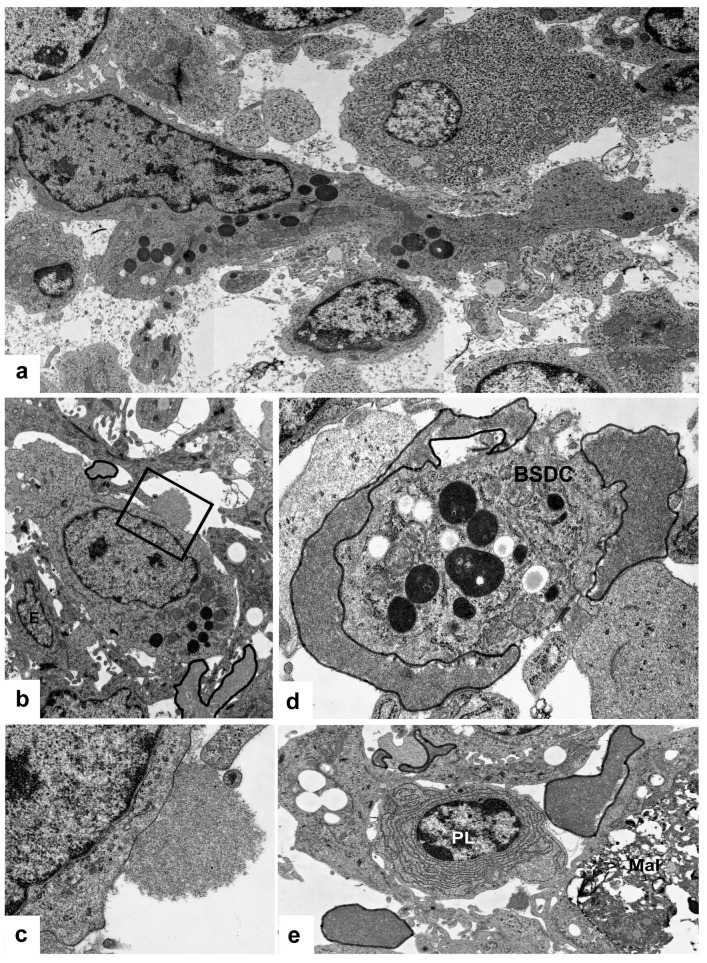
(**a**) Transmission micrograph of a BSDC. The eccentrically located nucleus and the cytoplasmic granules endow with polarity to the cell. There is a fine-flocculated substance around the cell. (**b**) Vaccinated bird. Masses of solid, extracellular substance attaches to a BSDC (outlined). No fine-flocculated substance around the BSDC. (**c**) Higher magnification from Figure 3b. (**d**) A BSDC process is covered by a solid mass of extracellular substance (outlined). (**e**) Inside the medulla, plasma cells (PL) may be formed. The outlined areas show aggregated substance.

**Figure 4 viruses-15-01301-f004:**
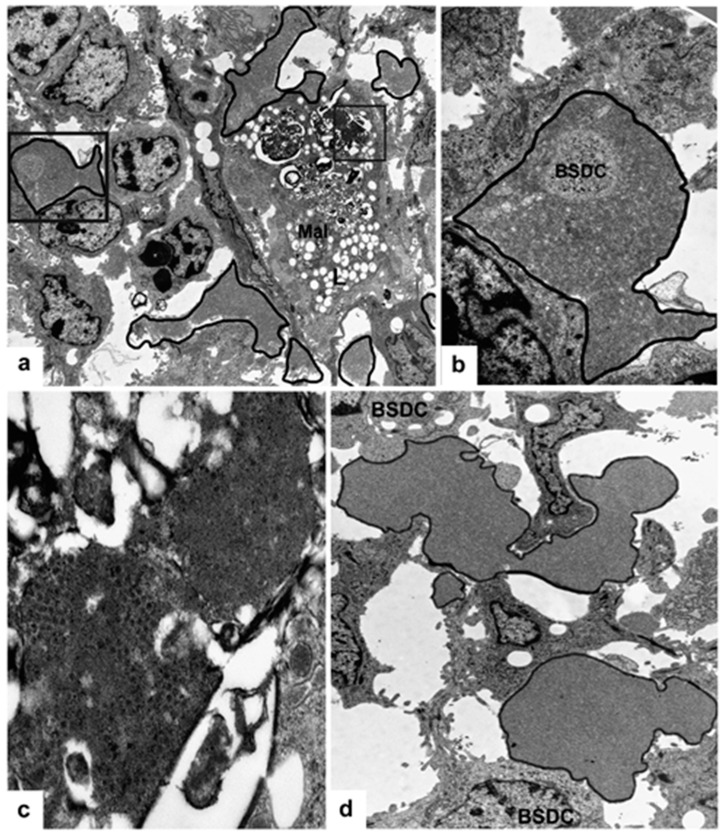
(**a**) Vaccinated bird. Mal with virus particles and numerous lipid droplets (L). The outlined areas show non-solubilized, solid, extracellular product of BSDC. The rectangular areas are shown on figures (**b**,**c**). (**b**) A BSDC process is firmly surrounded by solid, extracellular substance. (**c**) In the Mal, the virus particles are embedded in an electron-dense substance (**d**) In the depleted follicular medulla, the aggregated extracellular product of BSDC forms large irregular-shaped bodies (outlined).

**Figure 5 viruses-15-01301-f005:**
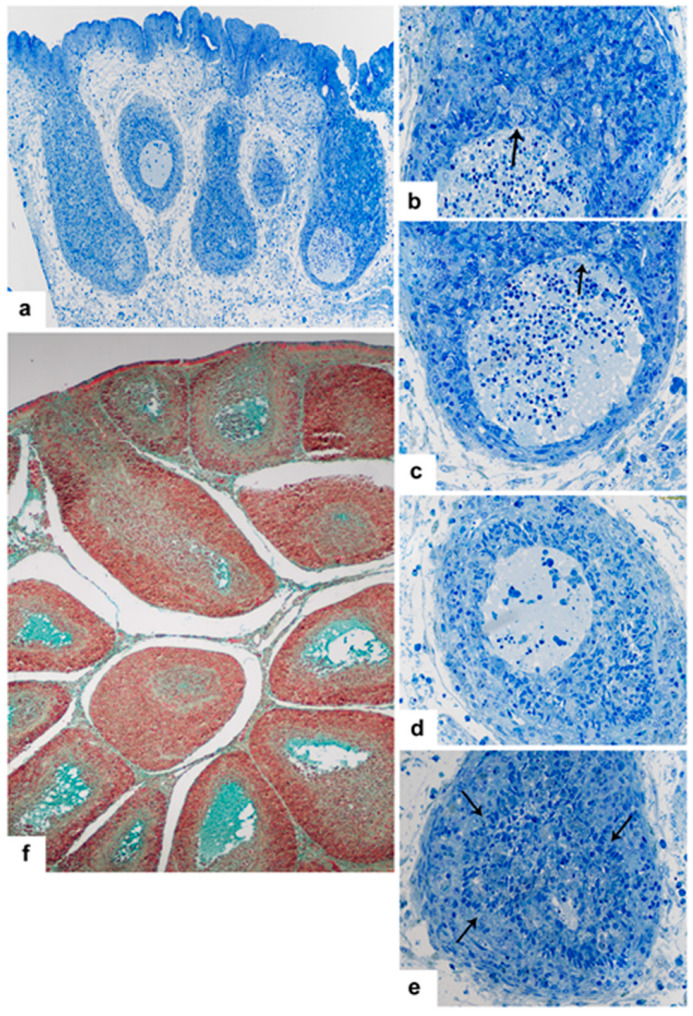
vvIBDV infected bird. Toluidine blue stained semithin (1.5 µm) sections (**a**–**e**). Four days post-infection, “cyst” formation takes place in the medulla. (**a**) The surface epithelium is wavy and in the medulla, “cysts” emerged. (**b**) The medullary “cyst” formation is created by fusion of Mal(s) (arrow). (**c**) The “cyst” has many apoptotic cells, Mals join to the “cyst” (arrow). (**d**) The cell number in the “cyst” dramatically decreased. (**e**) The gp from the “cyst” is absorbed, and the “cyst” collapsed. The cells of CMEA form a continuous layer around the medulla (arrows). (**f**) The Movat staining validates that the “cysts” contain gp.

**Figure 6 viruses-15-01301-f006:**
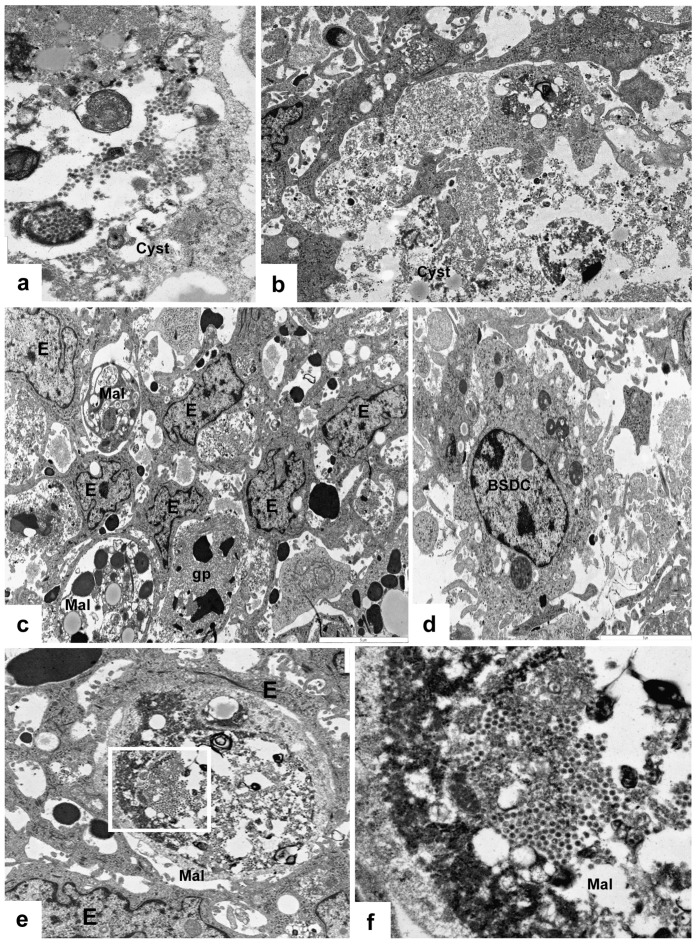
vvIBDV infected bird. (**a**) In addition to the free virus particles the granules of Mal also enclose viruses. (**b**) Other part of a “cyst” contains only cell detritus and a few free virus particles. (**c**) Wall of a “cyst” is formed by stellate-shaped epithelial cells and Mal(s). Around the “cyst”, many solid, electron-dense bodies occur. (**d**) Immature BSDC in the “cyst”. (**e**) In close proximity of a “cyst”, a Mal is “isolated” by epithelial cells (E). The rectangular area is shown on (**f**). The dark, electron-dense area is the remaining fused BSDC granules. The virus particles are embedded in the granular substance, which is possibly eliminated together with the viruses.

**Figure 7 viruses-15-01301-f007:**
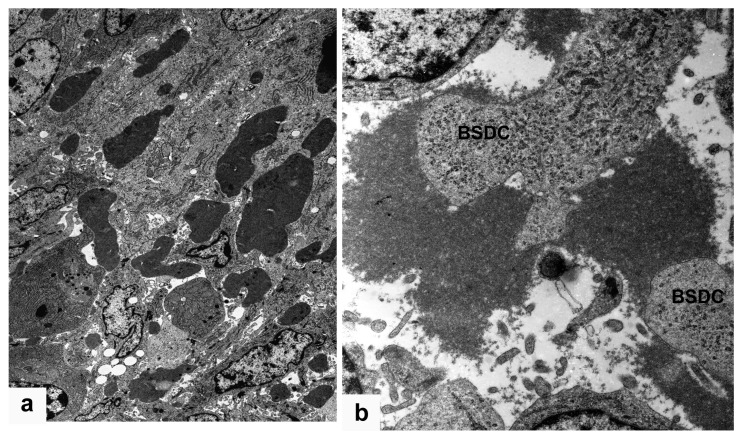
vvIBDV infected bird 21 dpi. (**a**) Electron-dense, solidified, extracellular bodies among the cells of FAE. (**b**) The aggregated substance surrounds a BSDC process, like in vaccinated chicken (see Figure 3d and Figure 4b).

## Data Availability

The data presented in this study are available in Oláh, I.; Felföldi, B.; Benyeda, Z.; Nagy, N.; Magyar, A.; Szőcs, E.; Soós, Á. Effect of Intermediate plus Vaccine and vvIBDV on Bursa Secretory Cells and Their Glycoprotein Production. Viruses 2023, 15, x.

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
