# Peer review of "Effect of Intermediate Plus Vaccine and vvIBDV on Bursa Secretory Cells and Their Glycoprotein Production"

_viruses, 2023, doi:10.3390/v15061301_

Round 1

Reviewer 1 Report (Previous Reviewer 2)

Comments and Suggestions for Authors

The revise MS entitled" Effect of vaccination and infection of bursal disease virus (IBDV) on the secretory cells and their glycoprotein production" was significantly approved. I thought it was worth a merit of publication after some trivial grammar problems were corrected. 

Lines 200-201: .............because the gp "held" the ch86 apoptotic receptor, and protects the B cell from nascent apoptosis....should be corrected as "holds"....

Author Response

Reviewer 1:

Thank you for your positive answer!

The trivial grammar problem were corrected. 

Reviewer 2 Report (Previous Reviewer 1)

Comments and Suggestions for Authors

The article give the impression of a summary of your previous article published earlier in different journals (poultry sciences , viruses etc..)

Author Response

Reviewer 2: 

The recent publication is not a summary of our previous papers, because this paper clearly shows the effect of vaccination and infection of IBDV for the bursal structure. It may be a short summary of secretory function of bursa of Fabricius.

Thank you for your opinion!

Round 2

Reviewer 2 Report (Previous Reviewer 1)

Comments and Suggestions for Authors

1) Title :

I would suggest  " Effect of intermediate plus vaccine (228E) and  vvIBDV  on bursa secretory cells and their glycoproteins production".

2) Materials and methods and results

As you know, vvIBDV belongs to genotype 3 and variant Delaware E strain belongs to genotype 2 (which is less pathogenic than genotype 3), however in your results there is no information on the results obtained for infection with l Delaware E strain ?. 

3) References

Serval references must be standarized 

Author Response

This manuscript is a resubmission of an earlier submission. The following is a list of the peer review reports and author responses from that submission.

Round 1

Reviewer 1 Report

Comments and Suggestions for Authors

According to the outstanding expertise of the authors on this subject,  It will be very interesting if this article could be a  review on secretory functions of bursa of Fabricius.

Other comments related to the submitted article:

Is there any reason for choosing only 3 days post vaccination for  IBDV challenge. 

You did use two different pathotypes of IBDV ( Very virulent IBDV and the US variant IBDV) is there any dIfference between the two viruses on the effect on bursa. because clinical signs between these viruses are different?

If you do consider that bursa of vaccinated groups is less  affected during post challenge, I think this is very interesting results which  illustrate an  early protection afforded by IBD vaccines . this aspect must be pointed out.

You did use chronic stage and acute stage. What do you mean in term of days ( from which day post challenge  we consider chronic stage?). 

Reviewer 2 Report

Comments and Suggestions for Authors

This manuscript described the secretory functions of chicken bursa of Fabricius. The content of this study seems to be novel and certainly can fill out some gaps about the pathogenesis of infectious bursa disease virus. However, in my opinion, the writing style of this MS was not sufficient to meet the standard for publication in "viruses". Therefore, I suggested the authors need an extensive editing before any positive response can be granted.

Specific comments were as follows:

1. The title of this MS needs to be re-considered again. It seemed to me that "effect of vaccination and infection of IBDV on the secretory cells and their glycoprotein production" may be more appropriate.

2. line 43: do not use "we" in the beginning of any sentence.

3. line 61: please define "gp" since it was the first time to show up in the MS.

4. lines 71-75: you need rephrase this paragraph. No need to mention the historic background about why you do this study.

5. lines 115-116: I did not understand what do the author mean "the number of experimental animals is not given"?

6. VvIBDV in several figure legends should be corrected as vvIBDV.

7. lines 278-279: I did not know what this sentence mean herein? You need condense this conclusions without any unnecessary explanation. 

8. If possible, please try to shorten your MS because it really looked like a dissertation.